# FLEX: Feature Importance from Layered Counterfactual Explanations

Anonymous Full Paper
Submission 42

## Abstract

Machine learning models achieve state-of-the-art performance across domains, yet their lack of interpretability limits safe deployment in high-stakes settings. Counterfactual explanations are widely used to provide actionable "what-if" recourse, but they typically remain instance-specific and do not quantify which features systematically drive outcome changes within coherent regions of the feature space or across an entire dataset.

We introduce **FLEX** (Feature importance from Layered counterfactual EXplanations), a model- and domain-agnostic framework that converts sets of counterfactuals into feature change frequency scores at *local*, *regional*, and *global* levels. FLEX generalises local change-frequency measures by aggregating across instances and neighbourhoods, offering interpretable rankings that reflect how often each feature must change to flip predictions. The framework is compatible with different counterfactual generation methods, allowing users to emphasise characteristics such as sparsity, feasibility, or actionability, thereby tailoring the derived feature importances to practical constraints.

We evaluate FLEX on two contrasting tabular tasks: traffic accident severity prediction and loan approval, and compare FLEX to SHAP- and LIME-derived feature importance values. Results show that (i) FLEX's global rankings correlate with SHAP while surfacing additional drivers, and (ii) regional analyses reveal context-specific factors that global summaries miss. FLEX thus bridges the gap between local recourse and global attribution, supporting transparent and intervention-oriented decision-making in risk-sensitive applications.

## 1 Introduction

Black-box models such as deep neural networks and ensemble methods regularly surpass traditional approaches across diverse tasks. However, their opacity raises concerns around trust, accountability, and intervention design in domains where errors can have severe consequences, such as traffic safety [1] or loan allocation.

Explainable AI (XAI) methods aim to mitigate this opacity. Among them, *counterfactual explanations (CFs)* are especially intuitive, as they ask: "What minimal changes to input features would alter the model's prediction?" Formally, given $X$ with prediction $a(X)$, a counterfactual $X'$ satisfies $a(X') \neq a(X)$ while typically aiming to minimise $||X' - X||$ [2]. This makes CFs well-suited for recourse: suggesting feasible changes to flip an undesirable outcome.

Yet counterfactuals alone do not indicate which features are *most important*, nor whether their importance is stable across regions of the data space. A feature critical in one subgroup may be irrelevant elsewhere. For example, in accident severity models, junction type may dominate predictions in one setting while weather plays a larger role in another. Similarly, in loan allocation, income stability may globally matter most, while education level is decisive in certain applicant groups.

We address this by introducing **FLEX**, a framework that extends counterfactual reasoning to derive feature importance on three levels:

- **Local**: feature changes for individual CFs.

- **Regional**: importance within a neighbourhood of similar instances.

- **Global**: importance aggregated across the dataset.

The main contributions are:

1. A generalisable method for deriving feature importance from counterfactuals which is agnostic to the counterfactual generation approach.

2. Regional and global analyses that highlight consistencies and divergences across the feature space.

3. Demonstrations across two case studies: traffic accident severity and loan allocation, compared to existing feature importance methods, showing FLEX's utility in identifying both domain-wide drivers and context-sensitive factors.

## 2 Related work

A growing line of work explores deriving feature importance from counterfactuals rather than just attributions. Meulemeester et al. [3] introduce *Counterfactual Feature Importance (CFI)* and a Shapley-style variant (CounterShapley) to quantify

**Figure 1.** Pipeline to calculate the FLEX score which involves having a factual (or set of factuals), generating counterfactuals, and observing feature changes.

per-feature influence along factual→counterfactual transitions. Kommiya Mothilal et al. [4] propose methods to assess necessity and sufficiency of features based on whether they change across a set of counterfactuals. DisCERN [5] generates case-based counterfactuals by borrowing values from nearest unlike neighbours, aiming to minimise actionable changes.

The DiCE framework [6] operationalises change-frequency-based importance both locally and globally: it computes local importance by measuring how often each feature changes across multiple counterfactuals for an individual instance, then aggregates these to yield global importance scores across a set of instances. However, DiCE does not provide a notion of regional importance, i.e., feature-change frequency aggregated over meaningful subgroups or neighbourhoods of similar instances. Additionally, all non-zero changes in continuous features above a small precision tolerance are treated equally.

FLEX builds on this foundation by explicitly formalising regional analysis and providing a user-defined threshold for capturing changes in continuous feature magnitude. It extends DiCE's frequency-based importance to multiple levels and provides regional–global correlation diagnostics to highlight where feature importance aligns or diverges across subsets. This enables context-sensitive insights, identifying features that drive outcome changes globally, as well as those that are particularly actionable within specific subpopulations. Furthermore, FLEX is agnostic to the counterfactual generation method: it can quantify feature importance using counterfactuals optimised for different desiderata such as sparsity [2], feasibility and actionability [7, 8], diversity [6], and could be extended to counterfactuals that enforce causal validity [9].

## 3 FLEX Method

Feature change frequency can serve as a key metric for evaluating the importance of features in counterfactual explanations. It is defined as the proportion of times a feature is changed between factual instances (undesirable predictions) and identified counterfactual instances (desirable predictions). The change frequency of feature $j$ of an instance $i$ for $N_{\mathrm{CF}}$ counterfactual samples is given by:

$$f_j^{instance}(i) = \frac{1}{N_{CF}} \sum_{k=1}^{N_{CF}} I\left(X_j(i,k) \neq X_j(i,\mathrm{orig})\right)$$
(1)

where

$$I\left(X_j(i,k) \neq X_j(i,\mathrm{orig})\right) = \begin{cases} 1, & \text{if } X_j(i,k) \neq X_j(i,\mathrm{orig}) \\ 0, & \text{otherwise} \end{cases}$$
(2)

For the $i^{th}$ sample, let the original value of feature $X_j$ be denoted as $X_j(i, orig)$, and let the value in the $k^{th}$ counterfactual instance of the $i^{th}$ sample be denoted as $X_j(i, k)$. By comparing $X_j(i, k)$ with $X_j(i, orig)$, using an indicator function $I(\cdot)$ that returns 1 if the values differ and 0 otherwise as shown in Eqn 2, it is recorded whether the feature changes in each counterfactual instance. The hypothesis is that if a feature is frequently modified in counterfactual instances, it may play an important role in altering the model's prediction; conversely, if a feature is seldom changed, the feature may not have a large influence on the classification outcome. FLEX can be applied to find both global and regional feature importances (Fig 2) as described in the next section as well as in Algorithm A.1. As shown, FLEX is agnostic to the counterfactual generation approach and can quantify feature importance using counterfactuals optimised for different desiderata.

### 3.1 Global and Regional Feature Change Frequency

In the global feature importance, the overall modification frequency of each feature is determined by statistically analyzing counterfactual instances generated for each sample across the dataset, thereby identifying the key features that consistently drive prediction changes. Regional feature importance is found by choosing a query factual and identifying the most similar samples to the query factual to form a region and then generating counterfactuals for all the samples in the region. A nearest neighbor algorithm using the Hamming distance is used to identify the samples most similar to the given query sample, where Hamming distance is defined as:

$$d_h(x,y) = \sum_{j=1}^{m} I(x_j \neq y_j)$$
(3)

Within this target region, the modification frequency of each feature is computed from the counterfactual instances, and the average and standard deviation of these local modification frequencies are further calculated. The global and regional feature change frequency can be calculated using a generic formula of $F_j$ for feature $X_j$ using the following formula:

$$F_j = \frac{1}{N_F} \sum_{i=1}^{N_F} f_j^{instance}(i)$$
(4)

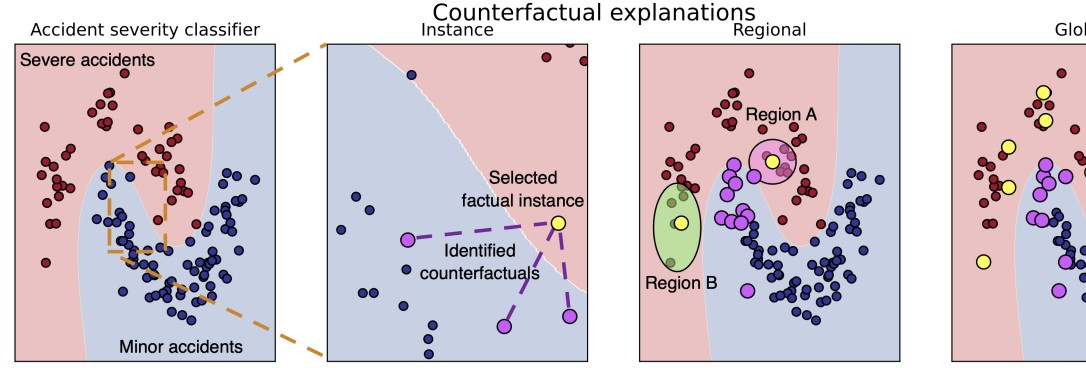

**Figure 2.** Demonstration on a toy 2D dataset labelled as the accident severity task. (1) Train a binary classifier for accident severity. (2) Select high-severity instances (yellow) and their nearby low-severity counterfactuals (lilac, $N_{\text{cf}} = 3$). (3) Regional case: select nearest neighbours around a point (green/pink regions) and their counterfactuals (lilac). (4) Global insights: randomly sample high-severity points across the space and identify counterfactuals. Feature change frequency is computed to quantify differences between factuals and counterfactuals

where global frequency $F_j^{global}$, and regional frequency, $F_j^{region}$, may use different values for $N_F$ and $N_{\text{CF}}$ given the differing magnitude of datapoints they each represent.

## 3.2 Assessing Regional Feature Distribution Changes

To assess how individual features behave within a given sample and its corresponding regional group compared to the generated counterfactuals, we employ a mode-based statistical method.

First, for each feature, we identify the most common category (i.e., the mode) and calculate its proportion within the factual data points. This provides a baseline for the primary distribution of that feature. We then analyze how this distribution changes in the corresponding counterfactual instances by calculating the proportion of the same mode within the counterfactual data.

By comparing these proportions, we can observe whether and to what extent the mode's prevalence changes during counterfactual generation. This change is quantified using the relative change rate, defined as:

$$\delta_j = \frac{p_j^{cf} - p_j^{orig}}{p_j^{orig}} \tag{5}$$

where $p_j^{orig}$ is the proportion of the most common category for feature $j$ in the factual sample, and $p_j^{cf}$ denotes the proportion of the same category in the counterfactuals. The metric $\delta_j$ quantifies the percentage change in the proportion of that category during the counterfactual generation process and provides a standardized measure. If $\delta_j$ is positive, it indicates that the proportion of that category has increased in the counterfactual data; if negative, the proportion has decreased.

When $|\delta_j|$ is small, it indicates that the counterfactual generation process has little impact on the main

distribution of that feature, demonstrating that the local explanation is relatively stable and consistent with respect to that feature. Conversely, when $|\delta_j|$ is large, it suggests that significant shifts occur in that feature during the counterfactual generation process, which may imply that the feature plays a more critical role in the regional explanation.

## 3.3 Correlation Analysis of Regional and Global Change Frequencies

We calculate the correlation between the regional and global feature change frequencies in order to understand if different regions are affected predominantly by global behaviour or region-specific behaviour. The correlation between the regional $F_j^{region}$ and the global change frequency $F_j^{global}$, is calculated using the Pearson correlation coefficient (PCC) as follows:

$$r = \frac{\sum_{j=1}^{m}(F_j^{region} - \mu_{re})(F_j^{global} - \mu_g)}{\sqrt{\sum_{j=1}^{m}(F_j^{region} - \mu_{re})^2}\sqrt{\sum_{j=1}^{m}(F_j^{global} - \mu_g)^2}} \tag{6}$$

where $\mu_{re}$ and $\mu_g$ represents the mean regional and global feature change frequencies respectively.

In Fig 3, the example points A, B, C, and D are illustrate different feature correlation scenarios, where the line y = x, represents equal global and regional importance of a particular feature. Therefore, the distance of a data point from this line reflects the degree of consistency or deviation between its global and regional importance.

Point A represents data in the upper-left quadrant, where the feature exhibits low global feature importance but high regional feature importance. This indicates that while the feature does not significantly impact the overall model predictions, it is very important for defining the accident severity classification of samples in this data region.

Point B represents data in the lower-left quadrant, where the feature neither stands out on a global scale nor exhibits exceptional prominence regional, suggesting that its influence on model predictions remains low both globally and regionally.

Point C represents data in the upper-right quadrant, where the feature has high importance in both global and regional dimensions, signifying that it plays a crucial role not only in the overall dataset but also within specific localised samples.

Point D represents data in the lower-right quadrant, where the feature shows high global importance but is not prominent regionally, indicating that, although it has a significant impact on overall model predictions, its importance does not manifest in some individual samples or regions.

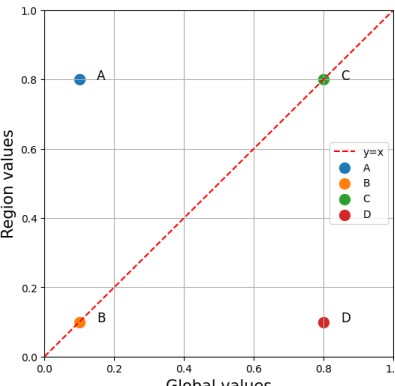

**Figure 3.** Example points A, B, C, and D are for illustration only: A indicates low global but high regional importance; B shows both are low; C indicates both are high; and D shows high global but low regional importance. The distance of each point from the line y = x reflects the consistency or deviation between global and regional importance.

## 4 Experimental

We evaluate FLEX on two datasets from distinct domains:

1. **Traffic accident severity** (Addis Ababa, Ethiopia, 2017–2020, [10]): 12,316 records, 11 categorical features.

2. **Loan allocation** (Kaggle[1]): 4269 records, continuous, categorical and ordinal features describing loan applicants.

Where appropriate, data was preprocessed by encoding text features and each dataset was then randomly divided into an 80% training set and a 20% test set. For each dataset, we train a random forest classifier and generate counterfactuals using DiCE

---

[1] https://www.kaggle.com/datasets/architsharma01/loan-approval-prediction-dataset/

[6]. Generalisability was assessed using 10 fold cross validation within the training set. Immutable features are excluded from CF changes (e.g., age, sex in traffic data). Global analysis uses random factual sampling and $N_F = 200$, and $N_{cf} = 10$ were utilised, meaning that 200 factual instances were randomly selected from across eachundesirable factual class (severe accident and loan rejection). Regional analysis selects groups based on contrasting values (e.g., high vs. low driving experience). For each region one factual instance which met the stated certain criteria was randomly selected and its four nearest neighbours which met the same selection criteria, resulting in $N_F = 5$ to form a region of five similar instances. As with the global approach, for each factual $N_{cf} = 10$ counterfactuals were identified for each. The average change frequency and standard deviation for each feature change were then calculated. Code implementing the technique for these experiments will be available upon publication[2].

## 5 Result and Discussion

### 5.1 Application 1: Traffic accident severity

Cross validation performance across 10 folds demonstrated mean (std) accuracy and F1 score performance of 83.44% (0.53) and 78.60% (0.55) respectively . The final model, trained on the combined training and validation sets, had accuracy and F1 score performance on the test set of 83.47% and 78.94%. This performance is comparable to other ML models that have accuracies between 70 to 85% and F1 scores between 77 to 86% [10]. Therefore we deemed our models to be sufficient for effective counterfactual generation. DiCE resulted in sparse feature changes in the generated counterfactuals, with the majority of counterfactuals changing two or fewer features values (1460 instances of 2000, 70.3%). Such concise explanations are both interpretable and practical, reflecting real-world constraints where only a few factors can reasonably be adjusted.

#### 5.1.1 Global Feature Change Frequency

We look at the global feature change frequency (where the value varies between 0 to 1) to identify which features were globally the most important for accident severity classification (Table 1).

There is a high level of agreement in the ranking of feature importance between FLEX and SHAP. FLEX provides the inherent advantage of measuring uncertainty with the standard deviation. The high change frequencies for *Driving experience*, *Cause of accident* and *Types of Junction* indicate

---

[2] https://github.com/anoynmous_repo_to_share

**Table 1.** Comparison of feature rankings by SHAP and FLEX (Rank & Score $\mu \pm \sigma$) for accident severity.

| Feature | FLEX | SHAP |
|---|---|---|
| Driving experience | 1 (0.34 ± 0.24) | 3 (0.0224) |
| Cause of accident | 2 (0.33 ± 0.23) | 2 (0.0267) |
| Type of Junction | 3 (0.29 ± 0.24) | 1 (0.0277) |
| Type of collision | 4 (0.28 ± 0.22) | 6 (0.0144) |
| Vehicle movement | 5 (0.23 ± 0.22) | 4 (0.0177) |
| Weather conditions | 6 (0.22 ± 0.22) | 7 (0.0107) |
| Pedestrian movement | 7 (0.22 ± 0.17) | 9 (0.0035) |
| Light conditions | 8 (0.20 ± 0.21) | 5 (0.0169) |
| Road surface type | 9 (0.10 ± 0.13) | 8 (0.0040) |

that these three features are adjusted most often to change a "severe" prediction to a "minor" accident severity prediction. This aligns with previous work, which found roadway design and speed-related factors dominant via SHAP [11], and another which flagged the importance of junction type [12]. The least importance feature was *Road surface type*, indicating that it is unlikely to make a large difference to the severity of the collision compared to the other features considered by the model. The high standard deviation of the values across all features suggest inconsistencies in the feature importance across samples and is highly dependent on the context and region we consider, therefore motivating a regional approach.

### 5.1.2 Regional Feature Change Frequency

In addition to considering the global importance of features across the entire data space, certain features may be important in a particular context for accident severity. Therefore, we explore the change frequency in local regions. We demonstrate this for regional approach for factuals chosen using contrasting feature categories for two features: the number of years of *Driving experience* (<1 year and >10 years) and *Weather conditions* ("normal" and "rainy"), resulting in four regions (Fig 4), which are compared to LIME-derived feature importances.

In Region 1 (Fig 4(a)), representing a sample of highly experienced drivers, a change in the *Type of Junction category* was the largest contributor to reducing accident severity. This differed from the importance features of LIME which places a greater importance on *Weather conditions*. The features suggested to be least important in influencing a reduction in accident severity were the *Road surface type*. In contrast, Region 2 (Fig 4(b)) represents a region of inexperienced drivers. For this region, the lack of driving experience itself was found to be the most important feature in changing the accident severity from severe to minor. The second most important feature was the *Weather conditions* and these both agreed with the values obtained from LIME. In contrast to the experienced drivers of Region 1, the *Vehicle movement* was consistently one of the least important features in reducing accident

severity for FLEX and *Cause of accident* was one of the least important features for both FLEX and LIME.

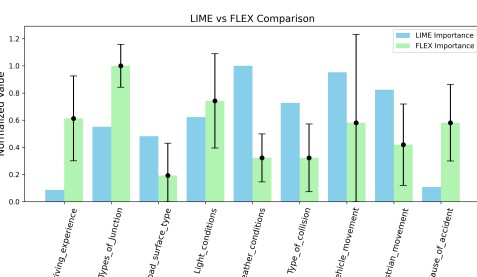

**(a)** Region 1: Driving experience >10 years

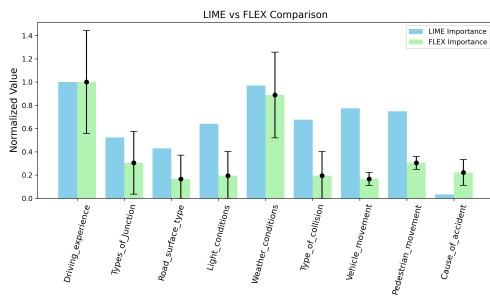

**(b)** Region 2: Driving experience <1 year

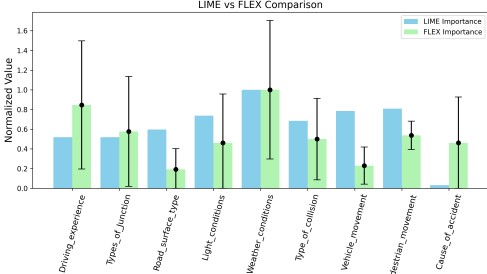

**(c)** Region 3: Weather conditions = "rainy"

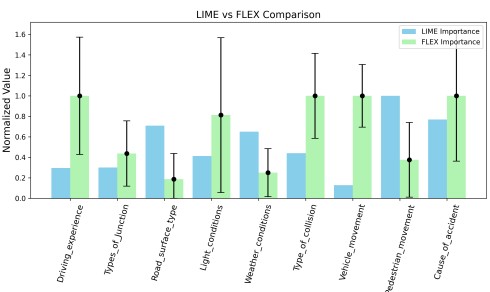

**(d)** Region 4: Weather conditions = "normal"

**Figure 4.** Mean and standard deviation of local feature change frequencies computed for four contrastive regions, each formed by randomly selecting a factual instance and its four nearest factual neighbours with a specific categorical feature value (for example, driving experience more than 10 years for region 1) and using 10 counterfactuals generated per factual instance. This is compared to importance scores from LIME which is normalized to be between 0 and 1.

Regions 3 and 4 demonstrate the approach for differing weather conditions: "rainy" and "normal" respectively. For the factual instances presented in Region 3 (Fig 4(c)), the *Weather conditions* themselves ("rainy") were the largest accident severity contributors, followed by features including the *Driving experience*: suggesting that sufficient driving experience is required in suboptimal weather conditions. Weather's elevated importance under rainy conditions echoes results from a study into older-pedestrian crashes, where precipitation amplified risk factors that were muted in global models [13]. In contrast, for Region 4 (Fig 4(d)), the "normal" weather conditions had a low impact on accident severity and the specific cause of the accidents were highly important in both FLEX and LIME.

These results reveal regional variation in the features that drive counterfactual changes in the accident severity. These differences reflect nuanced underlying variations in the feature distributions and local dynamics of each region, highlighting the value of regional analysis. For all four regions *Road surface type* was consistently one of the least important features, correlating with the global finding (Table 1). Similarities and differences between the regional and global importance are considered in more detail in a following section.

**Feature change insights for Region 1** To demonstrate the potential to obtain more in-depth insights into the reason for severe accidents, we have performed further analysis for Region 1 (high *Driving experience*) by considering the specific categorical feature values between factual and counterfactual instances. Table 2 compares the most common feature values in the original factual samples from Region 1 with their counterfactuals. The reduction in mode frequency for each feature is represented by $\delta_j$.

The high factual mode % for most features indicates that our regional approach successfully identified groups of similar accident situations. For this region, *Vehicle movement* was the largest contributor to accident severity. In the factual instances (severe accident severity), "Turnover" was the most frequent category, represented by 60% of factual instances, suggesting that this is a common cause of severe accidents. In the counterfactual instances, the frequency of "Turnover" dropped to 14% and instead the most common *Vehicle movement* category in the minor accident severity group was "Going straight".

A more modest drop in occurrence of the most frequent feature categories occurred for the other features and the mode in the counterfactuals remained the same as for the factual instances, suggesting that they were modified only for a small number of instances and that these feature changes were less

consistently important. The change in frequency of the driving experience for this group of experienced drivers (100% to 58%) is quite surprising as it suggests that a driver having less experience would lower accident severity. A possible reason could be overconfidence in driving ability. To address this, drivers could be retested several years after obtaining their license to prevent complacency. Further work is required, including involvement of domain experts, before drawing robust conclusions yet these initial findings reinforce the potential value of localised counterfactual analysis in uncovering key decision-driving factors within specific regions. The proposed method offers quantifiable insights that may be useful as part of wider analysis with domain experts.

**Table 2.** Comparison of the most frequent feature categories for the factual and counterfactual samples of Region 1. Factual Mode: the most frequent category for each feature in the original factual instances. Mode %: the proportion of factuals in that region with that categorical value. CF Mode % presents the proportion of counterfactual feature values populated by the factual mode. $\delta_j$: Relative change rate from equation 5.

| Feature | Factual Mode (%) | CF Mode % | $\delta_j$ |
|---|---|---|---|
| *Driving experience* | Above 10yr (100.00%) | 58.00% | -0.42 |
| *Types of Junction* | Crossing (100.00%) | 56.00% | -0.44 |
| *Road surface type* | Asphalt roads (100.00%) | 88.00% | -0.12 |
| *Light conditions* | Darkness – lights lit (80.00%) | 48.00% | -0.40 |
| *Weather conditions* | Normal (100.00%) | 86.00% | -0.14 |
| *Type of collision* | Vehicle with vehicle collision (100.00%) | 50.00% | -0.50 |
| *Vehicle movement* | Turnover (60.00%) | 14.00% | -0.77 |
| *Pedestrian movement* | Not a Pedestrian (100.00%) | 70.00% | -0.30 |
| *Cause of accident* | No distancing (40.00%) | 36.00% | -0.10 |

### 5.1.3 Comparing Regional and Global Feature Change Frequency

Figure 5 presents a comparison of global and regional feature change frequency for Regions 1 and 2. As outlined in Figure 3, this provides a means to assess alignment between global and local feature importance. Both of the regions show a weak to moderate correlation between global and regional feature change frequency: $r = 0.35$ and $r = 0.50$ respectively, indicating that counterfactual explanations are region-specific and influenced by local feature distributions, and that explanations based on global patterns may not generalize well across all data subsets.

The low regional and global impact of *Road surface type* (red circle on the plots) on accident severity is consistent between both the experienced and inexperienced drivers. Other than that, there are variations in which features align well with the global feature change frequency (near to the red dashed diagonal line) and those which stray from it. *Weather Conditions* and *Driving Experience* for Region 1, the experienced drivers, have a particularly high importance for this specific region and a modest global importance. For Region 1 all the other features are

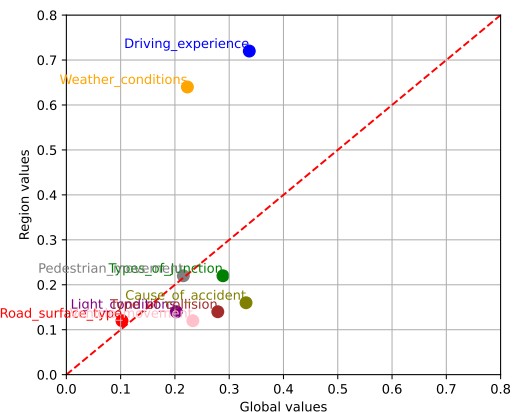

**(a)** Region 1: *Driving experience >10 years, r = 0.35*

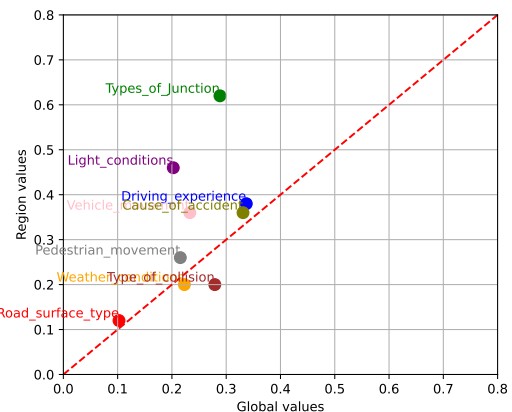

**(b)** Region 2: *Driving experience <1 year, r = 0.50*

**Figure 5.** Plotting the global feature change frequency values against those for each region. Pearson correlation coefficients ("r") are presented for each.

less important in this region than globally (below the dashed line), indicating that this group of accidents is quite different to the wider data set of road accident events. In comparison, Region 2 primarily has features above the dashed line, indicating increased importance compared to the global scenario.

These results underscore the importance of analyzing counterfactual behavior at both global and localized levels for more accurate and context-sensitive interpretability. Furthermore, this form of analysis could provide a complementary means to assess how similar a localised group of instances are to the wider data set, and to highlight similarities and differences to feature importances across a data set. Further work could include additional visualisation techniques to compare multiple regions.

## 5.2 Application 2: Loan allocation

The random forest classifier trained for this task achieved accuracy and F1 score performance on the test set of 98.13%. Both achieved the same performance because the weighted average was applied.

### 5.2.1 Global Feature Change Frequency

FLEX is a powerful algorithm for both categorical and continuous feature explainability. We observe in Table 3 how both SHAP and FLEX assign a high value to the cibil_score (credit score), solidifying it as the most globally essential feature in this analysis. FLEX produces close to same results as SHAP.

**Table 3.** Comparison of feature rankings by SHAP and FLEX (Rank & Score $\mu \pm \sigma$) for loan allocation.

| Feature | FLEX | SHAP |
|---|---|---|
| cibil_score | 1 (0.76 ± 0.24) | 1 (0.4053) |
| loan_term | 2 (0.20 ± 0.18) | 2 (0.0522) |
| loan_amount | 3 (0.16 ± 0.19) | 3 (0.0142) |
| luxury_assets_value | 4 (0.08 ± 0.12) | 5 (0.0065) |
| bank_asset_value | 4 (0.08 ± 0.12) | 8 (0.0046) |
| no_of_dependents | 4 (0.08 ± 0.10) | 9 (0.0027) |
| income_annum | 7 (0.06 ± 0.11) | 4 (0.0109) |
| commercial_assets_value | 8 (0.05 ± 0.12) | 7 (0.0051) |
| education | 8 (0.05 ± 0.11) | 10 (0.0027) |
| residential_assets_value | 10 (0.04 ± 0.08) | 6 (0.0058) |
| self_employed | 11 (0.03 ± 0.07) | 11 (0.0022) |

### 5.2.2 Effect of Threshold on Feature Importance

One important benefit of FLEX is the ability to compute importances based on a predefined threshold hyperparamater $\tau$ (see Algorithm A.1). We elucidate the effect of such an essential hyperparameter in Fig. 6, where we learn that the cibil_score changes frequently but not always by a large amount. This is a valuable insight that is not provided by alternative XAI approaches like SHAP or DiCE-derived feature importances (Algorithm A.2) [14]. Extensive comparative details can be found in Appendix A.2.

## 6 Conclusion and future work

A global and regional counterfactual explanation framework, FLEX, was developed. The framework was applied to two datasets, from different domains and composed of varying feature types.

FLEX revealed global insights into feature importance, with a clear ranking of overall importance which broadly aligned with SHAP rankings for both tasks. This happens despite FLEX being substantially more computationally efficient than SHAP (Appendix A.1). FLEX computes feature importances in $O(F \cdot N_s \cdot N_{cf})$, while (Kernel)SHAP is $O(2^F \cdot N_s)$, i.e., exponential in the number of features to explain $F$. Furthermore, FLEX returns standard deviations critical for assessing uncertainty, and offers the potential to tailor counterfactuals to specific desiderata: for example enforcing that the recourse path is feasible and actionable.

Change Threshold: 0.1

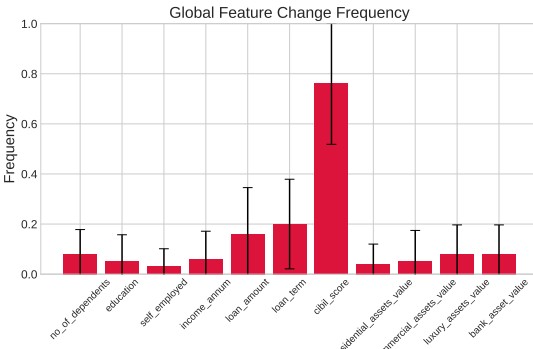

Change Threshold: 0.5

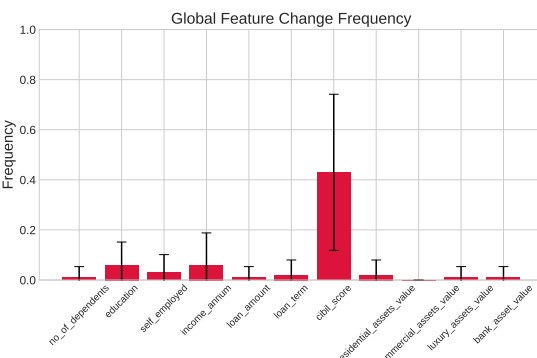

Change Threshold: 0.9

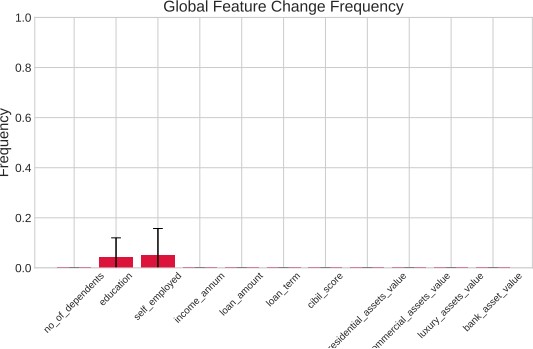

**Figure 6.** Impact of the threshold on loan allocation feature importances. (Top) $\tau = 0.1$, (Middle) $\tau = 0.5$, (Bottom) $\tau = 0.9$. Large thresholds naturally means detecting fewer changes. For instance the cibil_score changes frequently, but not always by a large amount.

Correlations between global and regional change frequencies for the accident severity task were at most r = 0.50, indicating that localised regional cases often deviate from wider trends.

In the loan allocation task, FLEX confirms that the cibil_score (credit score) is globally the most critical factor when allocating these loans. Analysis into the $\tau$ feature change magnitude hyperparameter confirmed that while cibil_score varies often, that change is not necessarily by a large amount. Such insights should prompt further investigations, and

can be used to make (or avoid making) decision that minimise (or perpetuate) the model's unfair biases.

Label encoding was used to convert any categorical feature values into numerical values, yet this introduces an artificial order. This can bias distance-based nearest-neighbour and counterfactual generation approaches, leading to inaccurate assessments of feature similarity and undermining the reliability of the explanations. As such, further work is required to refine the implementation for ordinal features and must include collaboration with domain experts. Utilising techniques to causal reasoning and building on recent work in this domain [15] would enable more robust conclusions to be drawn. Future work should also incorporate additional evaluation metrics for counterfactuals, especially those assessing sufficiency and necessity. Building on the framework from [4], these metrics would assess whether certain features are essential to maintain the original prediction (sufficiency) and how their modification affects the outcome (necessity), providing further insights.

The presented findings demonstrate that FLEX can clarify the decision-making process of black-box models and provide actionable insights alongisde input from domain experts and complimenting existing analytical techniques. By quantifying both global importance and regional variability, this approach enables more targeted interventions and insights to improve transparency in safety-critical tasks.

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

## A  Algorithms

### A.1  Comparison between FLEX and SHAP

The main FLEX algorithm, accounting for both categorical and continuous features, is presented in Algorithm A.1 with computational complexity $O(F \cdot N_s \cdot N_{cf})$ where $N_s$ is the number of factual instances, $F$ the number of features to vary, and $N_{cf}$ the number of counterfactuals per instance. With its linearity with respect to the number of features to vary, our counterfactual algorithm is a substantial improvement over the established (Kernel)SHAP and its exponential cost $O(2^F \cdot N_s)$ [16].

Additionally, A.1 covers both global and regional versions of FLEX. In the **global** case, the $N_s$ are generated randomly, while in the **regional** case, a first instance with a number of features of interest is chosen, then the remaining $1 - N_s$ are selected based on nearest neighbour search.

### A.2  Comparison between FLEX and DiCE

The primary distinctions between the FLEX algorithm (see A.1) and the DiCE global importance algorithm (see A.2) lie in the metrics they compute and their fundamental definition of what constitutes a significant feature change.

1. **Output Metrics:** The FLEX algorithm provides a more comprehensive, two-dimensional analysis by calculating two distinct metrics:

   - $\mathbf{\Phi}$: The frequency of feature changes, which indicates how often a feature is modified.
   - $\boldsymbol{\mu}$: The average relative magnitude of changes for continuous features, quantifying the typical size of an adjustment.

   In contrast, the DiCE algorithm (A.2) computes a single metric, $\mathbf{\Phi}^{\mathrm{DiCE}}$, which is equivalent to FLEX's frequency of change, but it does not measure the magnitude of these changes.

2. **Condition for a "Change" in Continuous Features:** A key conceptual difference is how each algorithm registers a change for a continuous variable.

   - In FLEX (A.1), a change is only counted towards the frequency score $\Phi_j$ if the normalized magnitude of the change exceeds a predefined threshold $\tau_j$ (i.e., $\frac{|x'_{k,j} - x_{q,j}|}{\mathrm{range}(X_j)} > \tau_j$). This allows the analysis to focus only on substantively significant alterations.
   - The DiCE algorithm (A.2), however, counts any non-zero modification to a feature's value as a change, as long as it is greater than a small machine precision tolerance $\epsilon$. It does not account for the significance or magnitude of the change.

---

**Algorithm A.1** FLEX - Counterfactual Feature Analysis

---

1: **procedure** ANALYZECHANGES($Q$, GenerateCFs, $N_{\text{cf}}$, $\boldsymbol{\tau}$, range)
    **Inputs:**
2:    $Q$: A set of $N_{\text{s}}$ query instances $\{\mathbf{x}_{q_1}, \ldots, \mathbf{x}_{q_{N_{\text{s}}}}\}$ for which the original outcome is undesirable.
3:    GenerateCFs($\mathbf{x}_q, N_{\text{cf}}$): A function that returns a set of $N_{\text{cf}}$ counterfactuals for an instance $\mathbf{x}_q$.
4:    $N_{\text{cf}}$: The number of counterfactuals to generate per instance.
5:    $\boldsymbol{\tau}$: A vector of change thresholds for continuous features, where $\tau_j$ is the threshold for feature $j$.
6:    range($X_j$): A function that returns the pre-computed range $(\max - \min)$ of a feature $j$ from the training data.

    **Outputs:**
7:    $\boldsymbol{\Phi}$: A vector of Global Feature Change Frequencies, where $\Phi_j$ is the frequency for feature $j$.
8:    $\boldsymbol{\mu}$: A vector of Global Average Relative Magnitudes for continuous features, where $\mu_j$ is the magnitude for feature $j$.

    **Initialization:**
9:    Initialize $\boldsymbol{\Phi} \leftarrow$ zero vector of size $d$
10:   Initialize $\boldsymbol{\mu} \leftarrow$ zero vector of size $d_{cont}$        ▷ $d$ is total features, $d_{cont}$ is continuous features
11:   Initialize two lists of lists, $L_\phi$ and $L_\mu$, to store per-feature per-instance results.

12:   **for** each instance $\mathbf{x}_q$ in $Q$ **do**
13:       $\mathcal{C}_q \leftarrow$ GenerateCFs($\mathbf{x}_q, N_{\text{cf}}$)        ▷ Generate counterfactuals
14:       **for** each feature $j = 1, \ldots, d$ **do**
15:          $c_j \leftarrow 0$        ▷ Change counter for frequency
16:          $m_j \leftarrow 0$        ▷ Accumulator for magnitude
17:          **for** each counterfactual $\mathbf{x}'_k$ in $\mathcal{C}_q$ **do**
18:             **if** feature $j$ is categorical **then**
19:                **if** $x'_{k,j} \neq x_{q,j}$ **then**
20:                   $c_j \leftarrow c_j + 1$
21:                **end if**
22:             **else if** feature $j$ is continuous **then**
23:                $\Delta_{k,j} \leftarrow |x'_{k,j} - x_{q,j}|$
24:                $m_j \leftarrow m_j + \frac{\Delta_{k,j}}{\text{range}(X_j)}$        ▷ Accumulate relative magnitude
25:                **if** $\frac{\Delta_{k,j}}{\text{range}(X_j)} > \tau_j$ **then**
26:                   $c_j \leftarrow c_j + 1$
27:                **end if**
28:             **end if**
29:          **end for**
30:          $\phi_j(\mathbf{x}_q) \leftarrow c_j/N_{\text{cf}}$        ▷ Calculate frequency change
31:          Add $\phi_j(\mathbf{x}_q)$ to list $L_\phi[j]$
32:          **if** feature $j$ is continuous **then**
33:             $\mu_j(\mathbf{x}_q) \leftarrow m_j/N_{\text{cf}}$        ▷ Calculate average relative magnitude
34:             Add $\mu_j(\mathbf{x}_q)$ to list $L_\mu[j]$
35:          **end if**
36:       **end for**
37:   **end for**

    **Aggregation:**
38:   **for** $j = 1, \ldots, d$ **do**
39:       $\Phi_j \leftarrow \frac{1}{N_{\text{s}}} \sum_{i=1}^{N_{\text{s}}} L_\phi[j][i]$        ▷ Average the frequencies
40:       **if** feature $j$ is continuous **then**
41:          $\mu_j \leftarrow \frac{1}{N_{\text{s}}} \sum_{i=1}^{N_{\text{s}}} L_\mu[j][i]$        ▷ Average the local relative magnitudes
42:       **end if**
43:   **end for**
44:   **return** $\boldsymbol{\Phi}, \boldsymbol{\mu}$
45: **end procedure**

---

---

**Algorithm A.2** DiCE - Global Feature Importance (reported and interpreted from [6])

---

1: **procedure** GLOBALIMPORTANCE($Q$, GenerateCFs, $N_{\text{cf}}$)
   **Inputs:**
2:     $Q$: A set of $N_{\text{s}}$ query instances $\{\mathbf{x}_{q_1}, \ldots, \mathbf{x}_{q_{N_{\text{s}}}}\}$ for which the original outcome is undesirable.
3:     GenerateCFs($\mathbf{x}_q, N_{\text{cf}}$): A function that returns a set of $N_{\text{cf}}$ counterfactuals for an instance $\mathbf{x}_q$.
4:     $N_{\text{cf}}$: The number of counterfactuals to generate per instance.

   **Outputs:**
5:     $\mathbf{\Phi}^{\text{DiCE}}$: A vector of Global Feature Change Frequencies, where $\Phi_j^{\text{DiCE}}$ is the frequency for feature $j$.

   **Initialization:**
6:     Initialize $\boldsymbol{C} \leftarrow$ zero vector of size $d$                           ▷ Global change counters for $d$ features
7:     Initialize $N_{\text{total\_cf}} \leftarrow 0$                                            ▷ Total number of counterfactuals generated

8:     **for** each instance $\mathbf{x}_q$ in $Q$ **do**
9:         $\mathcal{C}_q \leftarrow$ GenerateCFs($\mathbf{x}_q, N_{\text{cf}}$)                     ▷ Generate counterfactuals
10:        **for** each counterfactual $\mathbf{x}'_k$ in $\mathcal{C}_q$ **do**
11:            $N_{\text{total\_cf}} \leftarrow N_{\text{total\_cf}} + 1$
12:            **for** each feature $j = 1, \ldots, d$ **do**
13:                **if** feature $j$ is categorical **then**
14:                    **if** $x'_{k,j} \neq x_{q,j}$ **then**
15:                        $C_j \leftarrow C_j + 1$
16:                    **end if**
17:                **else if** feature $j$ is continuous **then**
18:                    **if** $|x'_{k,j} - x_{q,j}| > \epsilon$ **then**       ▷ $\epsilon$ is a small tolerance, e.g., $10^{-6}$
19:                        $C_j \leftarrow C_j + 1$
20:                    **end if**
21:                **end if**
22:            **end for**
23:        **end for**
24:    **end for**

   **Aggregation:**
25:    Initialize $\mathbf{\Phi}^{\text{DiCE}} \leftarrow$ zero vector of size $d$
26:    **if** $N_{\text{total\_cf}} > 0$ **then**
27:        **for** $j = 1, \ldots, d$ **do**
28:            $\Phi_j^{\text{DiCE}} \leftarrow C_j / N_{\text{total\_cf}}$           ▷ Normalize by total number of counterfactuals
29:        **end for**
30:    **end if**
31:    **return** $\mathbf{\Phi}^{\text{DiCE}}$
32: **end procedure**

---

