# OpenReview forum: "FLEX: Feature Importance from Layered Counterfactual Explanations"
_NLDL.org/2026/Conference — Submitted to NLDL 2026_

### Official Review · Reviewer_jYcN · 2025-10-01
**Region-based XAI is a promising angle, but contributions are not sufficient**

**Rating:** 2
**Confidence:** 4
**Final Rating:** 2
**Final Confidence:** 4

**Summary:**

The authors present a framework to aggregate counterfactual explanations on tabular data into input-regional (or global) averages. This helps analyze where counterfactual explanations differ across subsets of the data. One multi-step analysis on a traffic accident dataset and a smaller experiment on a loan allocation dataset have been conducted and show (1) some correlation of global-FLEX with SHAP, and (2) regional deviations from the global aggregate.

**Strengths:**

- The paper is fairly easy to read, the contribution and analyses are clear.
- The authors intend to make their code public upon paper publication.
- The major strong point of the paper is the analysis section of the traffic accident dataset, where several analysis steps, building upon each other, are presented, and some insights are obtained.
- Key related works are highlighted.
- Analyses and method description appear correct.

**Weaknesses:**

- The related works is weak overall, presenting only a handful of papers, and not including foundational literature of counterfactual explanations (going back at least until Wachter et al., 2017) or recent developments (2024+).
- The metric for distribution-change assessment (Sec. 3.2) is not fully justified. Why not use e.g. KL-divergence or other existing distribution divergence measures that capture the whole distribution, and not just the mode? The answer only becomes clear in Sec. 5.1.2, where it is not asked "how much" a feature has changed, but only the most frequent category is looked, at for the sake of interpretability.
- The methodological contribution of the paper appears to be mainly the averaging of an existing change-frequency metric over subsets (input-space neighbourhoods, or the whole dataset) of the data. While simplicity is good, this contribution may not be sufficient.
- It is not clear why, for one experiment, SHAP is used (for which sample-based values could be obtained, hence also a standard-deviation), and for another one LIME (which is notoriously unfaithful and may as well be replaced by SHAP).
- In section 5.1.3, the difference between FLEX and LIME is highlighted, but not further investigated, leaving me wondering what to make of this. (Can it be verified, maybe in a synthetic setting, which of them is more faithful/useful?)
- Section 5.2.1, does not add much to the paper. Instead, some quantitative evaluation across multiple datasets could strengthen the case for FLEX.
- Only 2 datasets have been used.
[Minor] In my opinion, section 3.3. could be much more concise, leaving out the explanation of the toy plot.

**Final Justification:**

While the paper is clear and easy to read, the methodological contribution is weak (evaluating an existing metric over data subsets) and not backed up by sufficiently thorough evaluations. While the authors promise more evaluations, their scope and outcome are unclear; they may or may not provide further justification for the proposed method.
Furthermore, even after the rebuttal, picking LIME as an evaluation tool over SHAP in some scenarios does not make sense to me.

**Justification:**

While the overall approach of region-based explainability is interesting, and parts of the traffic accident dataset analysis are insightful, the methodological contribution is weak (mainly averaging across data subsets). This could have been compensated for by an exceptional experimental section, demonstrating clear benefits that could not have been obtained otherwise, but this was not the case. The lackluster embedding of the work within the broader literature does not help either.

Despite the weaknesses, I encourage the authors to improve on their work, as I see the potential benefits of region-based explainability. Perhaps a real-world use case and quantitative multi-dataset evaluations would strengthen the paper.

---

> ### Author Rebuttal · Authors · 2025-10-22
>
> Thank you for taking the time to read and review our paper. We are glad to hear you found the paper easy to follow and insightful with aspects such as the contribution and the analysis being clear. Additionally, we appreciate the reviewer's feedback on the related work and we will expand on our current related work section to address the comment.
>
>
> > The metric for distribution-change assessment (Sec. 3.2) is not fully justified. Why not use e.g. KL-divergence or other existing distribution divergence measures that capture the whole distribution, and not just the mode? The answer only becomes clear in Sec. 5.1.2, where it is not asked ``how much'' a feature has changed, but only the most frequent category is looked, at for the sake of interpretability.
>
> We recognize that it is simplistic for the sake of interpretation and ease of introducing the technique, especially if we want the approach to be used for guided decision-making in areas such as road infrastructure. Though we do agree that divergence measures which capture the entire distribution can be useful and we plan on working on it for future work.
>
> > It is not clear why, for one experiment, SHAP is used (for which sample-based values could be obtained, hence also a standard-deviation), and for another one LIME (which is notoriously unfaithful and may as well be replaced by SHAP)
>
> We use SHAP and LIME as SHAP is generally used for global feature importances whilst LIME is used for local/regional feature importances. Moreover, by using both SHAP and LIME, it gives us more feature importance approaches to compare with FLEX which can enable us to better understand the behavior of FLEX.
>
>
> > In section 5.1.3, the difference between FLEX and LIME is highlighted, but not further investigated, leaving me wondering what to make of this. (Can it be verified, maybe in a synthetic setting, which of them is more faithful/useful?)
> We are happy to investigate further and demonstrate in synthetic dataset scenario.
>
> > Section 5.2.1, does not add much to the paper. Instead, some quantitative evaluation across multiple datasets could strengthen the case for FLEX.
>  We are Happy to add more evaluation and happy to implement established faithfulness metrics to assess FLEX and existing feature importance values. We've identified MoRF, LeRF, Fidelity [1], and $PGI^2$ [2] usable for the popular Red Wine Quality dataset [3].
>
> For additional quantitative evaluations, we also refer to our new experiment detailed in ``Explain how is the threshold selected, and how sensitive are rankings to it?'' to reviewer E9hH.
>
> > Only 2 datasets have been used. [Minor] In my opinion, section 3.3. could be much more concise, leaving out the explanation of the toy plot.
> We plan on adding an additional synthetic dataset to our other 2 real world datasets. Additionally,  We agree to make it more concise and potentially move certain content to the appendices if necessary to accommodate new content.
>
>
> - [1] Zheng et al. F-Fidelity: A Robust Framework for Faithfulness Evaluation of Explainable AI, ICLR, 2025
> - [2] Gajewski et al. Accurate Estimation of Feature Importance Faithfulness for Tree Models, AAAI, 2025
> - [3] Cortez et al. Modeling wine preferences by data mining from physicochemical properties, Decision Support Systems, 2009

---

### Official Review · Reviewer_wYv9 · 2025-10-03
**FLEX: Feature Importance from Layered Counterfactual Explanations**

**Rating:** 2
**Confidence:** 3

**Summary:**

This paper proposes a model- and domain-agnostic framework for explaining feature importance after training with various learning methods. The framework generalizes the idea of local change frequency in the generation of counterfactuals. Based on numerical experiments, the paper demonstrates that (i) its global ranking correlates closely with that of SHAP, and (ii) regional analyses reveal context-specific factors. In addition, the authors claim that FLEX requires significantly less computational effort compared to SHAP.

**Strengths:**

Although the underlying idea is similar to DiCE, one of the main strengths of the proposed method lies in its flexibility. It can be combined with any counterfactual generation technique and any type of learner. Furthermore, the proposed method requires less computational effort compared to conventional SHAP. Numerical experiments demonstrate its effectiveness.

**Weaknesses:**

Although the idea is simple and easy to understand, I think it would be better to clarify some of the notations and methods. For example, in Section 3.1 the Hamming distance is defined between sample elements, but nothing is mentioned about continuous features. Moreover, when using the nearest neighbor, the method essentially retrieves only one sample within the neighborhood. However, in Figure 2 it appears that multiple elements are collected in the neighborhood in the “Regional case.” This discrepancy should be clarified more carefully.

This issue contributes to my negative judgment of the paper.

In addition, the authors should carefully proofread the manuscript, as there are several mistakes. For example:

References [4] and [16] are duplicates.

p.2, l.181: $N_F$ is not defined.

p.3, l.236: “represents” $\to$ “represent”

p.3, l.238: “are illustrate” $\to$ “illustrate”

p.4, l.288: “eachundesirable” $\to$ “each undesirable”

p.5, l.340: “The least importance feature was” $\to$ “The least important feature was”

**Justification:**

Since the proposed method can be regarded as a variation of DiCE, it is important to make the original “regional” case precise and well-defined. Unfortunately, due to the unclear definition of distance for continuous features and the ambiguity regarding the use of nearest neighbors, the novelty of the contribution is not apparent in its current form. In addition, the quantity $\delta_j$ defined in (5) does not appear in Algorithm A.1, making it unclear how this value is actually used or reported in the paper.

To sum up, I believe this paper still has substantial room for improvement before it can be considered for publication.

---

> ### Author Rebuttal · Authors · 2025-10-22
>
> Thank you for taking the time to read and review our paper. We are glad to hear Reviewer wYv9 found the approach easy to understand, flexible and validated through numerical experiments.
> We are happy to clarify notations and methods, specifying aspects such as  continuous features.
>
> > Moreover, when using the nearest neighbor, the method essentially retrieves only one sample within the neighborhood. However, in Figure 2 it appears that multiple elements are collected in the neighborhood in the “Regional case.” This discrepancy should be clarified more carefully.
>
> The nearest neightbour is used to generate the region of interest. The idea is to capture region level behaviour by selecting the nearest neighrbours of a point o f interest to generate a region. After the region is generated (from a point and its k-nearest neighbours), counterfactuals are generated for each data point in the region to enable the FLEX score to be calculated. We will revise the manuscript to clarify how the nearest neighbour approach links to the overall method.
>
> > Add equation (5) to Algorithm 1?
>
> Thanks for pointing out typos. Algorithm 1 does not include (5) because that is a downstream metric specific to mode analysis of the features. Alg. 1 is meant to simply generate the feature importances, useful for such analyses. This clarification will be added to the camera ready version of our paper.

---

### Official Review · Reviewer_E9hH · 2025-10-09
**Paper introduces a counterfactual-based explainability method with promising results. I recommend acceptance.**

**Rating:** 4
**Confidence:** 3

**Summary:**

The paper introduces a model-agnostic framework FLEX (Feature Importance from Layered Counterfactual Explanations) that derives interpretable feature importance scores from sets of counterfactual explanations at three hierarchical levels: local (instance-specific), regional (within neighborhoods of similar instances), and global (across the dataset). FLEX quantifies how each feature must change to modify a model's prediction and refines this with magnitude thresholds for continuous variables, enabling more context-aware insights. Applications to traffic accident severity and loan approval show that FLEX's global rankings align with SHAP but reveal additional, region-specific drivers of predictions, which offers a scalable approach for understanding black-box models.

**Strengths:**

- The definition of feature change frequency and its aggregation are clear. FLEX links local, regional, and global interpretability. It can work with any counterfactual generator which increases generality.
- FLEX shows adaptability to categorical and mixed-type data (traffic and credit datasets).
- The paper discusses label encoding bias, ordinal feature handling, and reliance on CF generators.
- The paper is well-structured and easy to follow. The mathematical formulation is valid, and discussion about it is consistent. The pseudocode is provided.

**Weaknesses:**

- The dataset sample is small: only two tabular datasets.
- Label encoding for categorical features imposes order, which could impacts distance and frequency measures.
- No explanations of how results change with parameteres like $\tau$, neighborhood size, ..

**Justification:**

Summary:
The paper presents a framework (FLEX) for deriving interpretable feature importance from counterfactual reasoning. In general, the methodology is coherent, and clear. Empirical evaluation could be expanded, and some implementation details require clarification.

Remarks:
- In the introduction, discuss a bit more the meaning of feature importance, perhaps with an example
- In line 73, perhaps remove the word 'generalisable' as it is a bit vague
- Contribution 3 could be split into two contributions for more clarity
- In line 88, add a brief exaplanation of Counterfactual Feature Importance (CFI) and its Shapley-style variant (CounterShapley)

Questions:
- The paper argues FLEX helps interpret models, but how does it improve human understanding, or decision-making outcomes ?
- How is the user-defined threshold applied ? Is it per-feature, or applied globally?
- The definition checks if a feature changed or not. What about the influence of the magnitude of features ? For instance, if a "cost" feature changes from 10.000$ to 100.000$ ?
- Would there be a bias if we have many similar CFs ?
- Explain how is the $\tau$ threshold selected, and how sensitive are rankings to it?
- Is it possible to have runtime comparisons ?

---

> ### Author Rebuttal · Authors · 2025-10-22
>
> Thank you for taking the time to read and review our paper. We are glad to hear you found the paper well-structured and easy to follow, and we appreciate your feedback on our experimental evaluation. Thank you for your remarks on elaborating on the feature importances (in the introduction section), as well as areas to add more detail and to further improve clarity.
>
> > The paper argues FLEX helps interpret models, but how does it improve human understanding, or decision-making outcomes ?
>
> The approach helps improve decision making outcomes as it enables us to see what features are the most important for a certain outcome within a particular set of scenarios (e.g region of space). An example of this could be the traffic accident scenario where it can be seen that weather conditions was one of the most important factors that lead to accidents for inexperienced drivers. As a result of this, this can guide policy makers to make decisions such as improving aspects such as water drainage to lead to less rain on roads to prevent severe accidents.
>
>
> > How is the user-defined threshold applied?
>
> The threshold is defined globally. Since the change is normalised by their range (line 24 in Alg.1), the same value of $\tau$ is used across all features.
>
> > What about the influence of the magnitude of the features?
>
> Since the feature change $\Delta$ is normalised by the range of that feature (line 24 of Alg.1), the range of the feature is taken into account in the calculation. This is used to calculate the relative magnitude which is important, not the absolute magnitude. Furthermore, one can mathematically show that the $m_j$ we obtain is the same as we would obtain if the features were originally normalised using the min-max scaler.
>
> Finally, we repeated the loan allocation experiment, this time with all features pre-normalised from the start with a StandardScaler. This was done to ensure that the counterfactual generating algorithms also deals with normalised features. Our results do not change, confirming that the cibi\_score is indeed the most important feature for this problem.
>
>
> > Would there be a bias if we have many similar CFs
>
> The counterfatuals generated would be based on the counterfactual explanation generation technique and the underlying dataset. Therefore, there would not be bias present unless there was bias inherently within the dataset. We agree though that bias as a broader topic is important and bias induced by Counterfactual explanation generation could warrant further investigation
>
>
>
>
> > Explain how is the threshold selected, and how sensitive are rankings to it?
>
> Thank you for your insightful question. Given that our continuous features are normalized, $\tau$ is selected to be as small to be sensitive to small changes in a feature, while avoiding numerical instabilities. We agree that the selection of this hyperparameter was not sufficiently explained in the paper. The quantities that change with $\tau$ are the continuous features' FLEX score (their mean and standard deviation), and the feature importance rankings).
>
> We have conducted an additional experiment on loan allocation, performing a parameter sweep for 301 values of $\tau$ uniformly distributed between 0 and 1. They confirm our selection strategy, showing that the mean FLEX scores for the most important features are most obvious at small $\tau$. As $\tau$ increases, the differences in mean FLEX scores become less discernible, and the ranking less reliable.
>
> Interestingly, the standard deviations of the FLEX score for cibil\_score indicate an inverted U curve; the results are most uncertain around $\tau=0.35$. This coincides with the average change magnitude of that feature ($\Delta_{k,j}$ in line 23 of Alg. 1), thereby clarifying our original comment that the cibil\_score changes often, but not always by a large amount (at least not compared to binary features like self\_employed and education, see Fig. 6).
>
> For this problem, when focusing on continuous features, we emphasize the choice of $\tau$ on the left, closer to $0$. Indeed, the very small $\tau$ is similarly motivated for DiCE feature importance (Alg. 2) (it is called $\varepsilon$, on the order of $10^{-6}$). However, to focus our insights on categorical features (or possibly detect these features, which are independent of the threshold), a large $\tau$ closer to 1 is preferable.
>
> We will add the figures and the analysis corresponding to this experiment in the final version of the paper.
>
>
> > Is it possible to have runtime comparisons ?
>
> We appreciate this question, and we will include the runtime comparison results in the final version of the paper, along with an optimised version of our code.

---

### Meta-Review · Area_Chair_Z6Ga · 2025-10-30

**Recommendation:** Reject
**Confidence:** 3

**Metareview:**

The reviewers viewed the hierarchical (local, regional, global) CF-based analysis as promising and intuitive. The framework’s flexibility (CF-generator-agnostic) and computational efficiency over SHAP were noted strengths, and the traffic accident case study effectively demonstrated regional insights.

However, two reviewers (wYv9, jYcN) raised major, overlapping concerns about the paper’s originality and rigor. The methodological advance appears incremental compared to existing work (e.g., DiCE), and the empirical evaluation is limited (only two datasets, with weak baselines; SHAP can also be applied at local/regional levels instead of LIME).

The authors’ rebuttal, which promises additional experiments (sensitivity analysis, new datasets, faithfulness metrics), confirms that the submission was incomplete. Therefore, I'm leaning towards rejecting this paper and encouraging the authors to include the proposed experiments and a stronger baseline (e.g., aggregated SHAP) for a future submission.

---

### Decision · Program_Chairs · 2025-11-05

**Decision:**

Reject

**Comment:**

Based on the reviewers and AC comments, the paper cannot be presented at the conference.